# Vitamin D Supplementation and Physical Activity of Young Soccer Players during High-Intensity Training

**DOI:** 10.3390/nu11020349

**Published:** 2019-02-06

**Authors:** Maria Skalska, Pantelis Theo Nikolaidis, Beat Knechtle, Thomas Johannes Rosemann, Łukasz Radzimiński, Joanna Jastrzębska, Mariusz Kaczmarczyk, Artur Myśliwiec, Paul Dragos, Guillermo F. López-Sánchez, Zbigniew Jastrzębski

**Affiliations:** 1Department of Pediatrics, Diabetology and Endocrinology, University Clinical Centre in Gdansk, 80-210 Gdansk, Poland; mariajastrzebska@hotmail.com; 2Laboratory of Exercise Testing, Hellenic Air Force Academy, 13671 Dekelia, Greece; pademil@hotmail.com; 3Institute of General Practice and for Health Services Research, University of Zurich, 8091 Zurich, Switzerland; thomas.rosemann@usz.ch; 4Department of Health Promotion, Gdansk University of Physical Education and Sport, 80-336 Gdańsk, Poland; lukasz.radziminski@wp.pl (Ł.R.); mkariush@gmail.com (M.K.); admysliwiec@wp.pl (A.M.); zb.jastrzebski@op.pl (Z.J.); 5Department of Pediatrics, Diabetology and Endocrinology, Gdansk Medical University, 80-210 Gdansk, Poland; joanna.jastrzebska@hotmail.com; 6Department of Physical Education, Sport and Physical Therapy, University of Oradea, 410087 Oradea, Romania; dpaul@uoradea.ro; 7Faculty of Sport Sciences, University of Murcia, 30720 Murcia, Spain; gfls@um.es

**Keywords:** soccer, training load, time motion, youth athletes

## Abstract

The aim of this study was to confirm that vitamin D supplementation of young soccer players during eight-week high-intensity training would have a significant effect on their motion activity. The subjects were divided into two groups: the experimental one, which was supplemented with vitamin D (SG, *n* = 20), and the placebo group (PG, *n* = 16), which was not supplemented with vitamin D. All the players were subjected to the same soccer training, described as High-Intensity Interval Training (HIIT). The data of the vitamin D status, time motion parameters and heart rate were collected just before and after the intervention. A significant increase in 25(OH)D concentration (119%) was observed in the supplemented group, while the non-supplemented group showed a decrease of 8.4%. Based on the obtained results, it was found that physical activity indicators in the players were significantly improved during small-sided games at the last stage of the experiment. However, taking into account the effect of supplementation with vitamin D, there were no statistically significant differences between the placebo and the supplemented groups; thus, the effect size of the conducted experiment was trivial.

## 1. Introduction

Currently, the role of vitamin D for the human body is increasingly being explored. In recent years, numerous scientific centers have demonstrated its importance for the human metabolism in their research. Vitamin D is vital for immunological processes, anti-inflammatory, anti-thrombotic blood, preventing cell death, etc. [1]. In scientific experiments, it has been proved that vitamin D can be highly effective in counteracting many diseases and can be involved in improving the effectiveness of pharmacological treatment [2]. These studies have shown, among other things, that the functioning of the nervous and muscular systems in humans could be highly dependent on the concentration of 25(OH)D in the blood [3]. Its insufficiency is very common in the populations of the Northern European countries. However, Constantini et al. [4] and Vierucci et al. [5] reported that in young and adult athletes from Southern European countries, vitamin D deficiencies have also occurred and could cause them to have physiological dysfunctions, or to lower their physical activity. Scientific data on 25(OH)D concentration or supplementation of vitamin D among athletes and its effect on exercise capacity or the functions of certain organs are not always consistent. Presumably, this is the effect of research on people of different nationalities and geographical criteria, athletes from various sports disciplines that are at different levels of training, and people of different ages and levels of physical activity. 

Research conducted by Girgis et al. [6], Stockton et al. [7], Wyon et al. [8], and Close et al. [9] showed that vitamin D is important for skeletal muscle function. Furthermore, Koundourakis et al. [10] showed that this was also the case for the nervous system; in particular, in the relation between the muscular system and nervous system. 

Stockton et al. [7] found high effectiveness of vitamin D supplementation on muscle strength in judo contenders, rowers, hockey players, track and field athletes, and dancers. Girgis et al. [6] studied the effect of eight-month vitamin D supplementation on physical capacity, muscle efficacy and changes in the insulin level in physically active subjects. Relatively few studies have focused on young, healthy, and physically active subjects. Previous research considered the effect of vitamin D on the work of skeletal muscles [6,7,11], bone metabolism [12], and injury risk [13,14,15] and recovery [16]. Tenforde et al. [17] claimed that vitamin D was necessary for muscles to function optimally and to prevent bones from overuse fracture, a frequent risk for competitive athletes. Common injuries in young athletes could be the result of high (25(OH)D deficiency related to physical activity, as well as limited exposure to solar radiation, especially in countries with low solar radiation in the winter [18].

Considering the effectiveness of playing soccer, Koundourakis et al. [10] came to interesting conclusions. He found a correlation between the 25(OH)D level and the efficiency of the neuro-muscular system in professional soccer players. What is more, Valtuena et al. [19] found correlations in a group of 536 female subjects from the same countries between muscle strength of upper limbs and the 25(OH)D level. However, different results were presented by Książek et al. [20], who found no significant relationship between muscle strength in professional soccer players from Poland tested during the winter, the season with the highest 25(OH)D deficiency. It must be stated, however, that the effect of vitamin D supplementation in the players during the season of its deficiency was not studied, and the test was performed only once. The need for vitamin D supplementation during and directly after the winter season in countries in northern Europe was suggested by Kopeć et al. [21]. They found a significantly lower level of 25(OH)D in the players during this time compared to the summer. 

On the basis of the literature presented above, it can be assumed that the majority of authors agree that the concentration of 25(OH)D in athletes has a significant and positive effect on the functioning of their nervous and muscular systems, which are of particular importance in strength and explosive exercises. Therefore, can we hypothesize that the concentration of 25(OH)D in soccer players may determine their physical activity during special training exercise such as small-sided games? The literature on the impact of small games on the effectiveness of soccer training is widely available. Some authors suggest that these training methods are the best means of training to improve the athletes’ effort and technical-tactical skills [22,23,24].

During small-sided games, players demonstrate their abilities with respect to coordination, strength and explosive as well as aerobic and anaerobic performance capabilities. Therefore, we can assume that players whose 25(OH)D concentration is high should be characterized by faster adaptation to specific physical efforts. Supplementation with vitamin D in those who are deficient in this vitamin is therefore advisable. It is suggested that the optimal dose of vitamin D per day is 4000 to 5000 IU, administered for at least two months [25]. The main purpose of our work was to demonstrate the effect of 8-week vitamin D supplementation on the increase in physical activity of young soccer players.

It has been hypothesized that supplementation with vitamin D may be one of the important factors supporting the process of training and contribute to physical, technical and tactical ability, as well as coordination improvement, exposed comprehensively during small-sided games.

## 2. Materials and Methods

The subjects and experimental procedures have been described in detail previously [26]. The study protocol was approved by Ethical Committee of the local Medical Association in Gdańsk (Nr KB–1/14) and the investigation was carried out following the rules of the Declaration of Helsinki of 1975. All players and their parents or legal guardians were provided with detailed information about the study procedures and gave their written consent form. Thirty-six top young junior soccer players, aged 17.5 ± 0.6 years, body mass 71.3 ± 6.9 kg, BMI 22.2 ± 1.8 kg/m^2^ were included in the placebo-controlled, double-blind study. The subjects were divided into two groups: the placebo one, *n* = 16 (that was subjected to HIIT only), and the experimental one, *n* = 20 (that was subjected to HIIT and vitamin D3 supplementation).

### 2.1. Training Programme

The experiment was conducted during an 8-week training cycle during the preparatory season in winter time (January–March). Participants, school dormitory residents nourished in the same way, were non-randomly allocated into either an experimental group (SG, *n* = 20, subjected to training and cholecalciferol supplementation) or a placebo group (PG, *n* = 16, subjected to training only). The selection to the groups was guided by the peak power (W/kg) using Wingate test [27] and PWC170 index (Kgm/kg/min) using PWC_170_ test [28,29] measures obtained prior to experiment. The training regimen was the same for all subjects and has been thoroughly described previously [26]. All the players were subjected to the same soccer training, described as High-Intensity Interval Training (HIIT). Small-sided games and interval run at anaerobic threshold (AnT) were performed on a field with a synthetic surface. The intensity of the effort was determined by heart rate (HR) that was equal or higher than AnT value but did not exceed 90% HRmax. Small-sided games were performed on the field size 32 × 22 m (3 vs. 3 on Tuesday) and 44 × 33 m (6 vs. 6 on Thursday) with 120 m^2^ of the surface per player. The subjects of both groups played 4 games, 4 min each, with a 3 min active break that involved march and muscle relaxing drills. On Fridays, the players performed 4 series, 5 min each of interval run with a 3 min active break (like during small-sided games). On the other days of the 1-week training cycle, the players were subjected to technical, tactical, speed and explosive strength drills. The participation in training sessions was 95%. 

### 2.2. Time Motion and Heart Rate Analyses

The distance covered within small-sided games was measured using previously validated [30,31] portable GPS devices (minimaxX version 4.0, Catapult Innovations, Melbourne, Australia) with a frequency of 10 Hz and analyzed using specialized software (Catapult Sprint 5.0, Catapult Innovations, 2010, Melbourne, Australia). During the games, the players wore vests with GPS devices placed on the upper back. As recommended in the instructions, the GPS devices were activated 15 min before starting the training session. 

Speed zones were divided individually for each player according to his maximal running speed (Smax) and running velocity at the lactate threshold (V/LT) [32]. Smax was determined using the same GPS device during a 30-m sprint. Previous research has shown that this distance is adequate to achieve maximal speed in adult athletes [33]. After the warm-up, the participants performed this sprint twice with 5 min of active recovery between sprints. The highest recorded speed value was considered the Smax.

We defined a sprint as a running velocity at 80% of Smax or higher, and HIR (High-Intensity Running) as a running velocity between V/LT and 80% Smax. These criteria ensure that the speed zones were assigned individually according to the potential of each player. Finally, the following speed zones were assumed: zone 1—standing/walking (0–1 m s^−1^), zone 2—walking/jogging (1–2 m s^−1^), zone 3—LIR (Low Intensity Running, 2 m s^−1^ ÷ V/LT), zone 4—HIR, V/LT—80% Smax), and zone 5—sprinting (>80% Smax).

In addition to the distance run by the player in each speed zone, the average distance and exercise intensity was calculated for four small-sided games. 

During the games, coaches motivated the players to increase their effectiveness. The HR responses were recorded in 5-s intervals using telemetry devices (Polar Team Sport System; Polar Electro OY, Kempele, Finland, 2013).

### 2.3. Supplementation

Subjects from the supplemented group were given the vitamin D bottle (Vigantol Merck), whereas the placebo group received identical bottles with sunflower oil. 

All participants were asked to take 10 droplets per day (around 5000 IU of vitamin D daily in the supplemented group) in the morning.

Before, during and after the experiment, the tested subjects lived in the school dormitory and were nourished in the same way. The diet of the players was standard, but included an increased amount of vegetables, fruit and dairy products (sports diet). One month before and during the experiment, the players did not take any vitamin or other sports supplements. 

### 2.4. Biochemical Analyses

Blood samples for assessment of 25 (OH)D were centrifuged 300 × g for 15 minutes at room temperature in order to receive blood plasma. A 4.9 mL S-Monovette tube with ethylenediaminetetraacetic acid (K 3 EDTA; 1.6 mg EDTA/mL blood) and separating gel (SARSTEDT AG & Co., Nümbrecht, Germany) were used. The analyses were performed immediately after the blood collection.

Plasma 25(OH)D concentration (range, 50–125 nmol/L) was carried out by the immunoenzymatic method with the final fluorescence reading at 450 nm (BIOMÈRIEUX, Marcy-l’Etoile, France; kit nr 30463) using mini Vidas analyzer before and after the training program. The manufacturer’s declared intra-assay CV of the method was 2.4%–6.4%, respectively, of the range.

### 2.5. Statistical Analyses

Data were analyzed using within-subject modeling (http://www.sportsci.org) and analysis of covariance (ANCOVA), with the pre-training values treated as covariates. Both pre-training measures and the training responses were compared using *t*-tests, an independent sample *t*-test for unequal variances for pre-training and a dependent sample *t*-test for the responses. In addition, the post- and pre-training difference (a change score) was calculated for each participant and the mean change scores were compared between SG and PG with the *t*-test for unequal variances and ANCOVA. To estimate the magnitude of the supplementation effect, the mean change score in non-supplemented individuals was subtracted from the mean change score in supplemented individuals. The difference in mean change score was then standardized with a pre-supplementation standard deviation calculated for all supplemented and non-supplemented individuals according to method proposed by Hopkins [34]. The difference in mean change and standardized difference were reported with 95% confidence limits. The magnitude of individual responses to training was as described by Hopkins [34]. SDIR, a measure of individual response was calculated as the square root of the difference between squares of the standard deviations of the change scores in the supplemented and control groups. The remaining analyses were performed using STATISTICA (version 12; StatSoft, Inc., 2014, www.statsoft.com). A *p* value < 0.05 was considered significant.

## 3. Results

Baseline level of 25-hydroxyvitamin D, 25(OH)D, did not differ between supplemented and non-supplemented groups (Table 1). Twenty-two subjects (61.1%) had baseline 25(OH)D level < 50 nmol/L (12 in the SG, and 10 in the PG). In the vitamin D-supplemented group (SG), the 25(OH)D increased from 48.5 ± 8.6 to 106.3 ± 26.6 nmol/L (p < 0.0001), whereas in the placebo group, it decreased from 47.5 ± 16.2 to 43.5 ± 16.9 nmol/L (*p* = 0.228).

Table 2 presents the initial (pre-training) and final (post-training) values of the physical activity of the players during small games in relation to the speed zones 1–5. With the exception of zone 4, i.e., the run with intensity above the anaerobic threshold, the players significantly improved their results. The highest physical activity was recorded in the trail below the anaerobic threshold (zone 2 and zone 3), i.e., walking (zone 2) and low-intensity running (zone 3).

Comparing the results of physical activity of soccer players during four small-sided games in the supplemented and un-supplemented groups, no significant differences were found in any of the analyzed indicators. However, the effect size determined according to the Hopkins method [34] was at the trivial level <0.2 or small 0.2–0.6 (Table 3).

## 4. Discussion

The basic aim of our work was to determine whether supplementation with vitamin D of young soccer players in the period of deficiency can have a significant impact on improving their physical activity during small-sided games after applying an 8-week high-intensity training. Based on the obtained results, it was found that the athletes significantly improved the indicators, which characterize their physical activity, at the end of the experiment. It can, therefore, be assumed that the training loads used in young soccer players were an effective stimulus to improve their locomotion abilities. However, taking into account the effect of supplementation with vitamin D, there were no significant differences between the placebo and supplemented groups, and the effect size of the conducted experiment was minor. Technical and tactical skills, as well as the athletes’ endurance, play a decisive role during small-sided games. Therefore, the effectiveness of soccer players during these games will depend to a large extent on the metabolic efficiency of the body (physical performance) and the efficiency of the muscular and nervous system. Movement coordination in relation to special (soccer) physical activity will be the relevant indicator [22,35]. In previous works, we have proved that supplementation with vitamin D of the same group of soccer players may show a positive trend with regard to improving the indicators characterizing their aerobic endurance [36], as well as speed and explosive power [26]. However, these results do not allow general conclusions to be drawn concerning the high effectiveness of vitamin D supplementation in soccer players in relation to their performance during the game. Therefore, we wanted to check whether such a relation exists with regard to the integral assessment of the locomotor skills of the soccer player performing key training forms, i.e., small-sided games. We did not find any works on this subject in the bibliography. As we have already suggested in the introduction, many authors have shown a positive effect of vitamin D supplementation on the muscular system [6,7,8,9]. On the other hand, Beaudart et al. [37] conducted a meta-analysis concerning the assessment of the effect of vitamin D supplementation on skeletal muscle function in athletes. The final conclusions involved 30 publications, which analyzed the effectiveness of vitamin D in athletes of various sports. 

Based on the studies included in this meta-analysis, vitamin supplementation has a small but positive impact on global muscle strength, more specifically on lower limbs. However, no impact was found on muscle mass and muscle power. These studies correlate with the conclusions of Koundourakisa et al. [10], that the effect of supplementation with vitamin D, in addition to muscle strength, is greater efficiency of the neuro-muscular function manifested in improving the maximum power and speed of locomotion of soccer players. Therefore, in our opinion, these motor skills, together with the complex physical act that is muscular coordination, can have a significant impact on playing soccer. Small-sided games are performed during training at various time intervals, usually two to five minutes and repeated two to six times [32]. Intensity of physical exertion during their performance oscillates between 80% and 90% HR_max_ [24]. Our research registered Pre-training 89.3 ± 4.40% HR_max_ vs. Post-training 90.4 ± 3.10 beats/minHR_max_.

Pre-training 25.9 ± 8.8% vs Post-training 23.6 ± 8.0%, presumably, was the effect of the distance run by the athletes in the high-intensity speed zone 4 (Table 2). In terms of physical activity, small-sided games include all forms of walking and running that are recorded during the match [23]. In our research, the players covered the greatest distance in zones 2, 3 and 5, i.e., walking and running low (LIR) and high intensity (HIR). Dellal et al. [38], when examining athletes playing in a similar 4 × 4 system, found that the total distance that professional players can cover in 4 minutes oscillates between 597 and 835 meters and depends mainly on the surface of the soccer pitch and the rules of the game. 

The results of the young soccer players examined are close to those values and were registered at the end of the preparatory period. In relation to the speed zones, the results of senior soccer players obtained by [32] using the same method of locomotion as in our research are similar, and in zone 3 (LIR) they were at the same level, 1175.9 ± 243.79 m, and in zone 4 (HIR) 416.0 ± 105.36 m. Therefore, looking for factors that improve the locomotion ability of soccer players seems to be justified and may concern, among other things, vitamin D supplementation. Taking into account the effect of vitamin D supplementation in our research, it can be noticed that the average increase in the distance covered in zone 4 (HIR) in the control group of soccer players was greater than in the placebo group 30.9 ± 188 m vs. −37.1 ± 172 m. Although the differences between the groups were not statistically significant, it can be assumed that the use of this vitamin dose (5000 IU per day) for 8 weeks and during its shortage could have a beneficial effect on the athletes’ effort capacity in the high intensity zone. Similar conclusions concern zone 3 (LIR) and 5 (Sprint), in which supplemented competitors run a greater distance than the competitors in the placebo groups. As we have already shown, the total distance covered by the players during small-sided games is the result of their effort, coordination and technical-tactical skills. In our studies, the average improvement of this rate before and after the experiment in the group of supplemented athletes was 33% higher than in the placebo group. The effect of improving the locomotion abilities of the athletes was accompanied by a decrease in HR_AT_, which may emphasize the suggestion that the athletes of this group have improved the ability to specific effort to a greater extent. It seems, therefore, that both indicators may suggest a beneficial effect of vitamin D supplementation on improving both exercise abilities as well as locomotion of soccer players (Table 3). Taking into account the research results of the soccer players’ performance during the 8-week high-intensity training; it can be assumed that it was an effective stimulus to improve their physical activity during small-sided games. In turn, when assessing the effect of vitamin D supplementation, no statistically significant differences in locomotion between control and placebo groups were observed. However, when examining the effect of the Hopkins [34] experiment, it was determined that the differences between the groups are at the trivial level (zone 4—high intensity) and low (total distance). Therefore, in our opinion, we should recommend supplementation with vitamin D to young soccer players in order to improve their specific physical abilities, especially in the period when deficits are observed. 

## Figures and Tables

**Table 1 nutrients-11-00349-t001:** Baseline parameters in the supplemented (SG) and non-supplemented (PG) groups.

Variable	SG (*n* = 20)	PG (*n* = 16)	*p* ^a^
25(OH)D (nmol/L)	48.5 ± 8.6	47.5 ± 16.2	0.804

SG—supplemented with vitamin D, PG—placebo group, a probabilities for the dependent t-test for paired samples.

**Table 2 nutrients-11-00349-t002:** Distance covered by soccer players in individual zones, percentage (%) of their share in relations to total distance, average distance (m) and intensity (HR) of effort in four small-sided games (SSG).

Variable	Pre-Training (*n* = 36)	Post-Training (*n* = 36)	Mean Change Score	*p* ^a^
Zone 1 (m)	174 ± 35	156 ± 37	−17 ± 35	0.005
Zone 2 (m)	563 ± 62	510 ± 82	−53 ± 82	0.0004
Zone 3 (m)	730 ± 202	961 ± 203	231 ± 200	<0.0001
Zone 4 (m)	531 ± 190	531 ± 214	0.7 ± 181.5	0.983
Zone 5 (m)	10.6 ± 17.0	23.5 ± 30.0	12.9 ± 21.8	0.001
Zone 1 (%)	9.2 ± 3.4	7.6 ± 3.0	−1.6 ± 2.1	0.00006
Zone 2 (%)	28.5 ± 4.7	24.1 ± 5.7	−4.4 ± 5.1	<0.00001
Zone 3 (%)	35.7 ± 8.0	43.5 ± 6.0	7.8 ± 7.2	−0.00001
Zone 4 (%)	25.9 ± 8.8	23.6 ± 8.0	−2.3 ± 6.9	0.054
Zone 5 (%)	0.53 ± 0.91	1.06 ± 1.35	0.53 ± 0.94	0.002
SSG (m)	499 ± 47	551 ± 60	52 ± 44	−0.00001
HR_SSG_ (beats/min)	173 ± 6	170 ± 6	−3 ± 6	0.006

^a^ probabilities for the dependent *t*-test for paired samples.

**Table 3 nutrients-11-00349-t003:** Mean change score according to group and the difference in mean change.

Variable	Group (Mean Change)		*p* ^a^	Difference in Mean Change (Standardized)
	>SG (*n* = 20)	PG (*n* = 16)		
Zone 1 (m)	−21.8 ± 26.3	−11.9 ± 43.1	0.429	−9.9, −35.2–15.5 (−0.28, −1.00–0.44)
Zone 2 (m)	−55.9 ± 95.3	−49.6 ± 65.0	0.816	−6.3, −60.8–48.2 (−0.10, −0.98–0.78)
Zone 3 (m)	234 ± 182	227 ± 228	0.918	7, −136–150 (0.04, −0.67–0.74)
Zone 4 (m)	30.9 ± 188	−37.1 ± 172	0.266	67.9, −54.2–190.0 (0.36, −0.29–1.00)
Zone 5 (m)	13.5 ± 23.2	12.2 ± 20.7	0.864	1.3, 13.6–16.1 (0.07, −0.80–0.95)
Zone 1 (%)	−1.9 ± 1.5	−1.3 ± 2.7	0.398	−0.7, −2.2–0.9 (−0.19, −0.65–0.27)
Zone 2 (%)	−4.6 ± 5.3	−4.2 ± 5.0	0.834	−0.4, −3.8–3.1 (−0.08, −0.81–0.66)
Zone 3 (%)	7.2 ± 6.0	8.6 ± 8.6	0.595	−1.4, −6.6–3.8 (−0.17, −0.82–0.48)
Zone 4 (%)	−1.4 ± 6.5	−3.5 ± 7.4	0.370	2.2, −2.7–7.0 (−0.30, −0.30–0.79)
Zone 5 (%)	0.55 ± 1.10	0.50 ± 0.73	0.871	0.1, −0.6–0.7 (0.05, −0.63–0.74)
SSG (m)	61.6 ± 40	41.2 ± 48.2	0.184	20.4, −10.3–51.1 (0.43, −0.22–1.09)
HR_SSG_ (beats/min)	−2.6 ± 5.5	−3.0 ± 6.2	0.855	0.4, −3.7–4.4 (0.06, −0.58–0.69)

^a^ the *t*-test for unequal variances; PG—placebo group, un-supplemented; SG—supplemented with vitamin D; for difference in mean change and standardized (with pre-training SD) difference in mean, 95% confidence limits are shown.

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
