# Peer review of "Vitamin D Supplementation and Physical Activity of Young Soccer Players during High-Intensity Training"

_nutrients, 2019, doi:10.3390/nu11020349_

Round 1

Reviewer 1 Report

It is an intriguing idea to consider that the seasonal variation in vitamin D status may have an effect on the ability of skeletal muscle to perform vigorous exercise.  This manuscript describes comparative exercise performances of two groups of young men, one of which received a daily oral dose of cholecalciferol in oil, the other a placebo of the oil vehicle.  The overall conclusion from the results was that there were no significant benefits on athletic performance of the group taking the vitamin D supplements.  This would be the likely outcome from the starting values of 25-hydroxycholecalciferol in blood plasma of around 48 nmol/L.  Although this is often defined as the bottom of the normal range of vitamin D status, there has never been any deleterious aspect of vitamin D function that has been identified in individuals with that level of 25-hydroxycholecalciferol in plasma.

Although it is concluded that there was no benefit of the vitamin D supplement on muscle performance, the authors indicated in Table 3 that there was a difference between the two groups in the Zone 4 exercise schedule.  It is puzzling however, to compare this with the Zone 4 schedule comparing all the subjects before and after a period of training in Table 2.  For all the exercise types, training showed a significant improvement, except for Zone 4.  It would be helpful if the authors were to discuss these different findings and provide some interpretation of the relevance of the two results to each other.

The final sentence (lines 275-277) are puzzling.  After demonstrating no real significant differences in the performance of the two groups it is recommended that athletes take a cholecalciferol supplement in winter to improve performance.  This recommendation does not follow logically from the preceding discussion and should be explained more clearly.

Other points:

1.            Throughout the text the term vitamin D is used to describe the substance being measured in blood plasma.  This is incorrect and implies that vitamin D, cholecalciferol, was actually being measured in plasma.  The substance being measured was 25-hydroxycholecalciferol, a very different molecule from cholecalciferol.

2.            Line 41  “shortages” would be better replaced with the word “insufficiency”

3.            Line 42  “proved” would be better replaced with the word “reported”

4.            Line 134: The abbreviation HIR does not seem to have been defined.  Is it “high intensity running”? If so, it should be defined.

5.            Line 159: “standard method” – the principal of the method should be stated. Was it for example an immunoassay or perhaps a competitive protein assay?

6.            Line 186, Table 1: Why is the non-supplemented group labelled CG here whereas in the Abstract and Table 3 this group is labelled PG?

Author Response

Comments and Suggestions for Authors

It is an intriguing idea to consider that the seasonal variation in vitamin D status may have an effect on the ability of skeletal muscle to perform vigorous exercise.  This manuscript describes comparative exercise performances of two groups of young men, one of which received a daily oral dose of cholecalciferol in oil, the other a placebo of the oil vehicle.  The overall conclusion from the results was that there were no significant benefits on athletic performance of the group taking the vitamin D supplements.  This would be the likely outcome from the starting values of 25-hydroxycholecalciferol in blood plasma of around 48 nmol/L.  Although this is often defined as the bottom of the normal range of vitamin D status, there has never been any deleterious aspect of vitamin D function that has been identified in individuals with that level of 25-hydroxycholecalciferol in plasma.

Answer: We agree with the expert reviewer and answer as follows: The aim of our project was to answer the question whether vitamin D can have a significant impact on the physical activity of young footballers during specialized training, such as small-sided games. They include both low and high intensity efforts. Therefore, the study of the effect of vitamin D on footballers' effort ability seemed especially important in the context of their training during the preparatory period. Such tests are particularly important in those groups of athletes who train in countries, where low concentration of 25(OH) D in blood is observed.

Although it is concluded that there was no benefit of the vitamin D supplement on muscle performance, the authors indicated in Table 3 that there was a difference between the two groups in the Zone 4 exercise schedule.  It is puzzling however, to compare this with the Zone 4 schedule comparing all the subjects before and after a period of training in Table 2.  For all the exercise types, training showed a significant improvement, except for Zone 4.  It would be helpful if the authors were to discuss these different findings and provide some interpretation of the relevance of the two results to each other.

Answer: We agree with the expert reviewer and specify as follows: The results of physical activity (locomotion) as given in table 2 and 3 relate to small-sided games (before and after the experiment). Zone 4 includes the locomotion (run) of athletes with intensity above the Anaerobic Threshold (HIT). Indeed, the lack of significant differences in running in this intensity zone is controversial. However, this situation can be explained by the fact that at the end of the preparatory period (second series of  the research), the players focus more on covering the distance by sprint and running at an intensity below the anaerobic threshold than on high intensity run. This behavior is most often observed during matches. Moreover, our research concerned the assessment of locomotion in real time during training and the players were not additionally motivated as during stress tests, hence the tactics implemented by players during small-sided games could also be an important factor conditioning the registered result. In our opinion, the fact that there is no significant improvement in the results in this zone does not diminish the effectiveness of the training loads, because the direction of preparations was shifted mostly towards sprints. It can be considered as more beneficial effect for footballers’ preparation. If these explanations are sufficient for the reviewer, after accepting them, we will put an appropriate explanation in the discussion.

The final sentence (lines 275-277) are puzzling.  After demonstrating no real significant differences in the performance of the two groups it is recommended that athletes take a cholecalciferol supplement in winter to improve performance.  This recommendation does not follow logically from the preceding discussion and should be explained more clearly.

 Answer: We agree with the expert reviewer and answer as follows: In our opinion, the recommendation of vitamin D supplementation is obvious for both - resting and exercise metabolism. As it was shown in the introduction, based on literature, vitamin D has a wide meaning in human life processes. Therefore, as we have shown in our research, despite the lack of its significant impact on the physical activity of young footballers, its supplementation, especially during periods of deficiency is indicated. It means it’s desirable during the winter period in countries where solar radiation is low. In addition, as we have already described in lines 271-274, despite the lack of significant differences in the measured parameters of locomotion between groups, we noticed a larger average increase in results in the supplemented group, eg in the intensity of LIT, HIR and Sprint. Therefore, a higher trend of improving the results, although not statistically significant, was observed in the supplemented group (290-294). Similar opinion is presented by Beaudart et. al. (214) who also suggest that vitamin D supplementation in athletes does not have to be spectacular so that it can be used and bring even a small improvement in athletic performance. This, in our opinion has a special importance in the groups of high-level athletes.

Other points:

Throughout the text the term vitamin D is used to describe the substance being measured in blood plasma.  This is incorrect and implies that vitamin D, cholecalciferol, was actually being measured in plasma.  The substance being measured was 25-hydroxycholecalciferol, a very different molecule from cholecalciferol.

Answer: We agree with the expert reviewer and answer as follows: 25(OH)D is an inactive form of vitamin D. Of course, its concentration is measured in blood plasma. It is referred to vitamin D storage. In our opinion, if we specifically write about the concentration of 25(OH)D in the blood plasma, this term should be used. If we are writing about supplementation, we should use the term vitamin D. Discussing the level of vitamin D in athletes, the term is used instead of 25(OH)D because they are convergent terms. In many of the publications cited in our article, this approach to terminology is compatible with our work.We will try to arrange this terminology according to this scheme and reviewer. If you prefer a different form, we are asking for suggestions.

Line 41  “shortages” would be better replaced with the word “insufficiency”

Answer: We agree with the expert reviewer and we have included this suggestion in the text.

Line 42  “proved” would be better replaced with the word “reported”

Answer: We agree with the expert reviewer and we have included this suggestion in the text.

Line 134: The abbreviation HIR does not seem to have been defined.  Is it “high intensity running”? If so, it should be defined.

Answer: We agree with the expert reviewer and in lines 140 and 144; we have defined the terms of HIR and LIR.

5.            Line 159 or 170?: “standard method” – the principal of the method should be stated. Was it for example an immunoassay or perhaps a competitive protein assay?

Answer: We agree with the expert reviewer and changed to ‘Plasma 25(OH)D concentration (range, 50-125 nmol/L) was carried out by the immunoenzymatic method with the final fluorescence reading at 450 nm (BIOMÈRIEUX, Marcy-I'Etoile, France; kit nr 30463) using mini Vidas analyzer before and after the training programme.

Line 170: The study was carried out using the immunoenzymatic method with the final fluorescence reading at 450 nm.

Answer: We agree with the expert reviewer and changed to ‘Plasma 25(OH)D concentration (range, 50-125 nmol/L) was carried out by the immunoenzymatic method with the final fluorescence reading at 450 nm (BIOMÈRIEUX, Marcy-I'Etoile, France; kit nr 30463) using mini Vidas analyzer before and after the training programme.

6.            Line 186, Table 1: Why is the non-supplemented group labelled CG here whereas in the Abstract and Table 3 this group is labelled PG?

Answer: We agree with the expert reviewer and we have included this suggestion in the text.

Reviewer 2 Report

In this study, the authors evaluated the effectiveness of Vitamin D supplementation & physical activity of young soccer players during high-intensity training, & concluded that vitamin D supplementation has no obvious benefit on athletic performance. It is an interesting observation. However, the manuscript needs to be modified to be consistent with the observations:

Despite no significant benefits with the supplement, it is not clear to the reviewer, why the authors are recommending supplements??

Throughout the text, the authors should be consistent, and use the term 25(OH)D was measured in serum.

It is not clear to the reviewer, why the authors chose not to measure the serum calcium & phosphate levels in their collected samples.

The authors should be careful in recommending & encouraging young soccer players with exogenous stimulants to enhance their performance.

Author Response

In this study, the authors evaluated the effectiveness of Vitamin D supplementation & physical activity of young soccer players during high-intensity training, & concluded that vitamin D supplementation has no obvious benefit on athletic performance. It is an interesting observation. However, the manuscript needs to be modified to be consistent with the observations:

Answer: We agree with the expert reviewer and we will try to explain this aspect in the further responses.

Despite no significant benefits with the supplement, it is not clear to the reviewer, why the authors are recommending supplements??

Answer: We agree with the expert reviewer and answer as follows: In our opinion, the recommendation of vitamin D supplementation is obvious in terms of both resting and exercise metabolism. As it was shown in the introduction, based on the literature, vitamin D is marked by a high importance in the physiological processes in a human body. Therefore, as we have shown in our research, its supplementation for young footballers is recommended, despite the fact that there is lack of its significant impact on their physical activity. Supplementation is favorable especially during winter period in the countries where the solar radiation is low, which leads to deficits of vitamin D. Moreover, as we have described in lines 271-274, despite the lack of significant differences in the measured parameters of locomotion between groups, we have registered a higher average increase in results in the supplemented group, e.g. in the intensity of LIT, HIR and Sprint. Therefore, a higher trend of improving the results, although not statistically significant, was observed in the supplemented group (290-294). Similar opinion has Beaudart et al. (214) who also suggests that vitamin D supplementation in athletes does not have to be spectacular to bring even a small improvement in the athletic performance. From our point of view, it has a particular significance in the groups of high-level sportsmen.

Throughout the text, the authors should be consistent, and use the term 25(OH)D was measured in serum.

Answer: We agree with the expert reviewer and answer as follows: 25 (OH) D is an inactive form of vitamin D and is described as being its storage. Of course, its concentration is assessed in the blood plasma. In our opinion, if we specifically write about the concentration of 25 (OH) D in the blood plasma, this term should be used. When we write about supplementation, we should use term vitamin D. Discussing the level of vitamin D in athletes, this term is used instead of 25 (OH) D because they are convergent terms.  This approach of terminology is consistent with many publications cited in our research. We will try to arrange this terminology according to this scheme. In case you prefer a different solution, we are open minded for your suggestions. 

It is not clear to the reviewer, why the authors chose not to measure the serum calcium & phosphate levels in their collected samples.

Answer: We agree with the reviewer's suggestion that the measurement of calcium and phosphorus levels is important during vitamin D supplementation. Of course, we have performed such tests for supplemented and un-supplemented players. Before as well as after the experiment, the level of these elements did not differ significantly in athletes and was within the reference limits. According to the literature, possible changes may eventually occur with long-term (more than 6 months) vitamin D supplementation at prominently higher values (over 10,000 IU / day). We did not describe these parameters because of the lack of adverse changes, but also because these issues did not concern the undertaken topic.

The authors should be careful in recommending & encouraging young soccer players with exogenous stimulants to enhance their performance.

Answer: We agree with the expert reviewer and answer as follows: Certainly, the use of any exogenous stimulants in young and even adult players should have medical justification. In case of our players it was an absolute recommendation because most of them had profound vitamin D deficiencies. In countries with low exposure to sunlight, this is widely reported. Moreover, diet, especially in the boarding school (95% of our players lived there) is poor with vitamin D, which additionally influences its low concentrations in our players. This is unfortunately a very common phenomenon in the countries of Northern Europe. For this reason, supplementation with vitamin D is recommended.